# You Can Have Better Graph Neural Networks by Not Training Weights at All: Finding Untrained GNNs Tickets

**Tianjin Huang[1], Tianlong Chen[2], Meng Fang[1,4], Vlado Menkovski[1],**
**Jiaxu Zhao[1], Lu Yin[1], Yulong Pei[1], Decebal Constantin Mocanu[1,3],**
**Zhangyang Wang[2], Mykola Pechenizkiy[1], Shiwei Liu[2]**

[1]Eindhoven University of Technology, [2]University of Texas at Austin
[3]University of Twente, [4]University of Liverpool

{t.huang,j.zhao,v.menkovski,l.yin,y.pei.1,m.pechenizkiy}@tue.nl
{tianlong.chen,atlaswang}@utexas.edu
{d.c.mocanu}@utwente.nl, {Meng.Fang}@liverpool.ac.uk
shiwei.liu@austin.utexas.edu

## Abstract

Recent works have impressively demonstrated that there exists a subnetwork in randomly initialized convolutional neural networks (CNNs) that can match the performance of the fully trained dense networks at initialization, without any optimization of the weights of the network (i.e., untrained networks). However, the presence of such untrained subnetworks in graph neural networks (GNNs) still remains mysterious. In this paper we carry out the first-of-its-kind exploration of discovering matching untrained GNNs. With sparsity as the core tool, we can find *untrained sparse subnetworks* at the initialization, that can match the performance of *fully trained dense* GNNs. Besides this already encouraging finding of comparable performance, we show that the found untrained subnetworks can substantially mitigate the GNN over-smoothing problem, hence becoming a powerful tool to enable deeper GNNs without bells and whistles. We also observe that such sparse untrained subnetworks have appealing performance in out-of-distribution detection and robustness of input perturbations. We evaluate our method across widely-used GNN architectures on various popular datasets including the Open Graph Benchmark (OGB).

## 1 Introduction

Graph Neural Networks (GNNs) [1, 2] have shown the power to learn representations from graph-structured data. Over the past decade, GNNs and their variants such as Graph Convolutional Networks (GCN) [3], Graph Isomorphism Networks (GIN) [4], Graph Attention Networks (GAT) [5] have been successfully applied to a wide range of scenarios, e.g., social analysis [6, 7], protein feature learning [8], traffic prediction [9], and recommendation systems [10]. In parallel, works on untrained networks [11, 12] surprisingly discover the presence of untrained subnetworks in CNNs that can already match the accuracy of their fully trained dense CNNs with their initial weights, without any weight update. In this paper, we attempt to explore discovering untrained sparse networks in GNNs by asking the following question:

*Is it possible to find a well-performing graph neural (sub-) network without any training of the model weights?*

Positive answers to this question will have significant impacts on the research field of GNNs. ① If the answer is yes, it will shed light on a new direction of obtaining performant GNNs, e.g., traditional training might not be indispensable towards performant GNNs. ② The existence of such performant subnetworks will extend the recently proposed untrained subnetwork techniques [11, 12]

T. Huang et al., You Can Have Better Graph Neural Networks by Not Training Weights at All: Finding Untrained GNNs Tickets. *Proceedings of the First Learning on Graphs Conference (LoG 2022)*, PMLR 198, Virtual Event, December 9–12, 2022.

**Figure 1:** Performance of untrained graph subnetworks (UGTs (ours) and Edge-Popup [12]) and the corresponding trained dense GNNs. We demonstrate that as the model size increases, UGTs is able to find an untrained subnetwork with its random initializations, that can match the performance of the corresponding fully-trained dense GNNs. The x-axis denotes the corresponding model size for each point, e.g. "64-2" represents a model with 2 layers and width 64.

in GNNs. Prior works [11–13] successfully find that randomly weighted full networks contain untrained subnetworks which perform well without ever modifying the weights, in convolutional neural networks (CNNs). However, the similar study has never been discussed for GNNs. While CNNs reasonably contain well-performing untrained subnetworks due to heavy over-parameterization, GNN models are usually much more compact, and it is unclear whether a performant subnetwork "should" still exist in GNNs.

Furthermore, we investigate the connection between untrained sparse networks and widely-known barriers in deep GNNs, such as over-smoothing. For instance, as analyzed in [14], by naively stacking many layers and adding non-linearity, the output features are prone to collapsing and becoming indistinguishable. Such undesirable properties significantly limit the power of deeper/wider GNNs, hindering the potential application of GNNs on large-scale graph datasets such as the latest Open Graph Benchmark (OGB) [15]. It is interesting to see what would happen for untrained graph neural networks. Note that the goal of sparsity in our paper is **not for efficiency**, but to obtain nontrivial predictive performance without training (a.k.a., "masking is training" [11]). We summarize our contributions as follows:

- We demonstrate for the first time that there exist untrained graph subnetworks with matching performance (referring to as good as the trained full networks), **within randomly initialized dense networks and without any model weight training**. Distinct from the popular lottery ticket hypothesis (LTH) [16, 17], neither the original dense networks nor the identified subnetworks need to be trained.

- We find that the gradual sparsification technique [18, 19] can be a stronger performance booster. Leveraging its global sparse variant [20], we propose our method – UGTs, which discovers matching untrained subnetworks within the dense GNNs at extremely high sparsities. For example, our method discovers untrained matching subnetworks with up to 99% sparsity. We validate it across various GNN architectures (GCN, GIN, GAT) on eight datasets, including the large-scale OGBN-ArXiv and OGBN-Products.

- We empirically show a surprising observation that our method significantly mitigates the over-smoothing problem without any additional tricks and can successfully scale GNNs up with negligible performance loss. Additionally, we show that UGTs also enjoys favorable performance on Out-of-Distribution (OOD) detection and robustness on different types of perturbations.

## 2   Related Work

**Graph Neural Networks.** Graph neural networks is a powerful deep learning approach for graph-structured data. Since proposed in [1], many variants of GNNs have been developed, e.g., GAT [5], GCN [3], GIN [4], GraphSage [21], SGC [22], and GAE [23]. More and more recent works point out that deeper GNN architectures potentially provide benefits to practical graph structures, e.g., molecules [8], point clouds [24], and meshes [25], as well as large-scale graph dataset OGB. However, training deep GNNs usually is a well-known challenge due to various difficulties such as gradient vanishing and over-smoothing problems [14, 26]. The existing approaches to address the above-mentioned problem can be categorized into three groups: (1) skip connection, e.g., Jumping

connections [27, 28], Residual connections [24], and Initial connections [29]; (2) graph normalization, e.g., PairNorm [26], NodeNorm [30]; (3) random dropping including DropNode [31] and DropEdge [32].

**Untrained Subnetworks.** Untrained subnetworks refer to the hypothesis that there exists a subnetwork in a randomly intialized neural network that can achieve almost the same accuracy as a fully trained neural network without weight update. [11] and [12] first demonstrate that randomly initialized CNNs contain subnetworks that achieve impressive performance without updating weights at all. [13] enhanced the performance of untrained subnetworks by iteratively reinitializing the weights that have been pruned. Besides the image classification task, some works also explore the power of untrained subnetworks in other domains, such as multi-tasks learning [33] and adversarial robustness [34].

Instead of proposing well-versed techniques to enable deep GNNs training, we explore the possibility of finding well-performing deeper graph subnetworks at initialization in the hope of avoiding the difficulties of building deep GNNs without model weight training.

## 3 Untrained GNNs Tickets

### 3.1 Preliminaries and Setups

**Notations.** We represent matrices by bold uppercase characters, e.g. $X$, vectors by bold lowercase characters, e.g. $x$, and scalars by normal lowercase characters, e.g. x. We denote the $i^{th}$ row of a matrix $A$ by $A[i,:]$, and the $(i,j)^{th}$ element of matrix $A$ by $A[i,j]$. We consider a graph $\mathcal{G} = \{\mathcal{V}, \mathcal{E}\}$ where $\mathcal{E}$ is a set of edges and $\mathcal{V}$ is a set of nodes. Let $g(A, X; \theta)$ be a graph neural network where $A \in \{0,1\}^{|V| \times |V|}$ is adjacency matrix for describing the overall graph topology, and $X$ denotes nodal features . $A[i,j] = 1$ denotes the edge between node $v_i$ and node $v_j$. Let $f(X; \theta)$ be a neural network with the weights $\theta$. $\| \cdot \|_0$ denotes the $L_0$ norm.

**Sparse Neural Networks.** Given a dense network $\theta_l \in \mathcal{R}^{d_l}$ with a dimension of $d_l$ in each layer $l \in \{1, ..., L\}$, binary mask $m_l \in \{0,1\}^{d_l}$ yielding a sparse Neural Networks with sparse weights $\theta_l \odot m_l$. The sparsity level is the fraction of the weights that are zero-valued, calculated as $s = 1 - \frac{\sum_l \|m_l\|_0}{\sum_l d_l}$.

**Graph Neural Networks.** GNNs denote a family of algorithms that extract structural information from graphs [35] and it is consisted of *Aggregate* and *Combine* operations. Usually, *Aggregate* is a function that aggregates messages from its neighbor nodes, and *Combine* is an update function that updates the representation of the current node. Formally, given the graph $\mathcal{G} = (A, X)$ with node set $\mathcal{V}$ and edge set $\mathcal{E}$, the $l$-th layer of a GNN is represented as follows:

$$a_v^l = Aggregate^l(\{h_u^{l-1} : \forall u \in \mathcal{N}(v)\}) \qquad (1)$$

$$h_v^l = Combine^l(h_v^{l-1}, a_v^l) \qquad (2)$$

where $a_v^l$ is the aggregated representation of the neighborhood for node $v$ and $\mathcal{N}(v)$ denotes the neighbor nodes set of the node $v$, and $h_v^l$ is the node representations at the $l$-th layer. After propagating through $L$ layers, we achieve the final node representations $h_v^L$ which can be applied to downstream node-level tasks, such as node classification, link prediction.

**Untrained Subnetworks.** Following the prior work [11], [12] proposed Edge-Popup which enables finding untrained subnetworks hidden in the a randomly initialized full network $f(\theta)$ by solving the following discrete optimization problem:

$$\min_{m(S) \in \{0,1\}^{|\theta|}} \mathcal{L}(f(X; \theta \odot m(S)), y) \qquad (3)$$

where $\mathcal{L}$ is task-dependent loss function; $\odot$ represents an element-wise multiplication; $y$ is the label for the input $X$ and $m$ is the binary mask that controls the sparsity level $s$. $S$ is the latent score behind the binary mask $m$ and it has the same dimension as $m$. To avoid confusion, here we use $m(S)$ instead of $m$ to indicate that $m$ is generated by $S$. We will use $m$ directly for brevity in the following content.

Different from the traditional training of deep neural networks, here the network weights are never updated, masks $m$ are instead generated to search for the optimal untrained subnetwork. In practice,

each mask $\boldsymbol{m}_i$ has a latent score variable $\boldsymbol{S}_i \in \mathcal{R}$ that represents the importance score of the corresponding weight $\boldsymbol{\theta}_i$. During training in the forward pass, the binary mask $\boldsymbol{m}$ is generated by setting top-$s$ smallest elements of $\boldsymbol{S}$ to 0 otherwise 1. In the backward pass, all the values in $\boldsymbol{S}$ will be updated with straight-through estimation [36]. At the end of the training, an untrained subnetwork can be found by the generated mask $\boldsymbol{m}$ according to the converged scores $\boldsymbol{S}$.

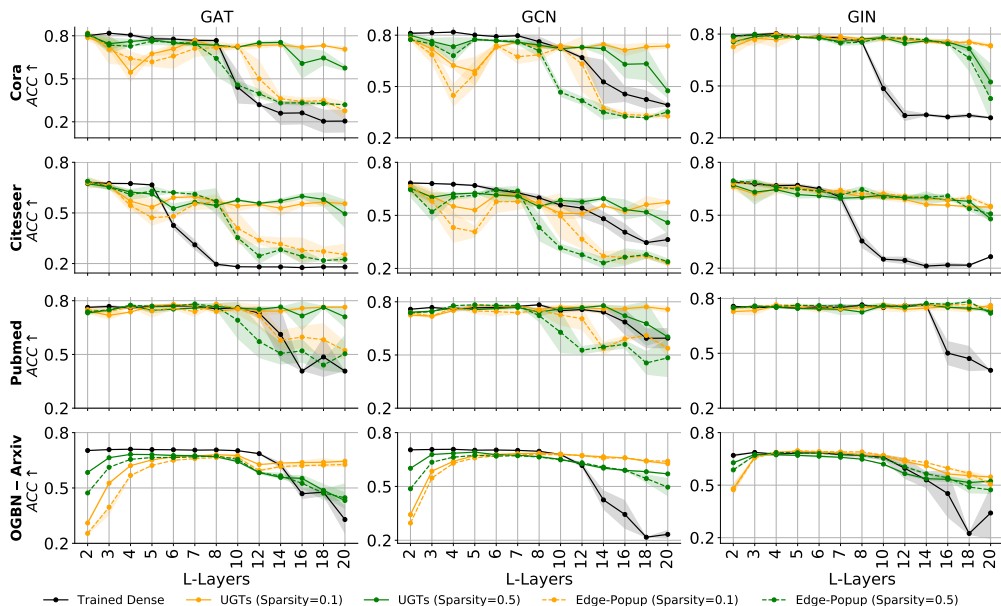

**Figure 2:** The performance of GNNs with increasing model depths. Experiments are conducted on various GNNs with Cora, Citeseer, Pubmed and OGBN-Arxiv. We observe that as the model goes deeper, fully-trained dense GNNs suffer from a sharp accuracy drop, while UGTs preserves the high accuracy. All the results reported are averaged from 5 runs.

### 3.2 Untrained GNNs Tickets – UGTs

In this section, we adopt the untrained subnetwork techniques to GNNs and introduce our new approach – Untrained GNNs Tickets (UGTs). We share the pseudocode of UGTs in the Appendix C.

Formally, given a graph neural network $g(\boldsymbol{A}, \boldsymbol{X}; \boldsymbol{\theta})$, where $\boldsymbol{A}$ and $\boldsymbol{X}$ are adjacency matrix and nodal features respectively. The optimization problem of finding an untrained subnetwork in GNNs can be therefore described as follows:

$$\min_{\boldsymbol{m} \in \{0,1\}^{|\boldsymbol{\theta}|}} \mathcal{L}(g(\boldsymbol{A}, \boldsymbol{X}; \boldsymbol{\theta} \odot \boldsymbol{m}), \boldsymbol{y}) \tag{4}$$

Although Edge-Popup [12] can find untrained subnetworks with proper predictive accuracy, its performance is still away from satisfactory. For instance, Edge-Popup can only obtain matching subnetworks at a relatively low sparsity i.e., 50%.

We highlight two limitations of the existing prior research. First of all, prior works [12, 13] **initially** set the sparsity level of $\boldsymbol{m}_i$ as $s$ and maintain it throughout the optimization process. This is very appealing for the scenarios of sparse training [37–39] that chases a better trade-off between performance and efficiency, since the fixed sparsity usually translates to fewer floating-point operations (FLOPs). This scheme, however, is not necessary and perhaps harmful to the finding of the smallest possible untrained subnetwork that still performs well. Particularly as shown in [20], larger searching space for sparse neural networks at the early optimization phase leads to better sparse solutions. The second limitation is that the existing methods sparsify networks layer-wise with a uniform sparsity ratio, which typically leads to inferior performance compared with the non-uniform layer-wise sparsity [20, 39, 40], especially for deep architectures [41].

**Untrained GNNs Tickets (UGTs)**. Leveraging the above-mentioned insights, we propose a new approach UGTs here which can discover matching untrained subnetworks with extremely high sparsity

**Table 1:** Test accuracy (%) of different training techniques. The experiments are based on GCN models with 16, 32 layers, respectively. Width is set to 448. See Appendix B.7 for GAT architecture. The results of the other methods are obtained from [42].

|  | Cora | | Citeseer | | Pubmed | |
|---|---|---|---|---|---|---|
| N-Layers | 16 | 32 | 16 | 32 | 16 | 32 |
| **Trained Dense GCN** | 21.4 | 21.2 | 19.5 | 20.2 | 39.1 | 38.7 |
| +Residual | 20.1 | 19.6 | 20.8 | 20.90 | 38.8 | 38.7 |
| +Jumping | 76.0 | 75.5 | 58.3 | 55.0 | 75.6 | 75.3 |
| +NodeNorm | 21.5 | 21.4 | 18.8 | 19.1 | 18.9 | 18 |
| +PairNorm | 55.7 | 17.7 | 27.4 | 20.6 | 71.3 | 61.5 |
| +DropNode | 27.6 | 27.6 | 21.8 | 22.1 | 40.3 | 40.3 |
| +DropEdge | 28.0 | 27.8 | 22.9 | 22.9 | 40.6 | 40.5 |
| **UGTs-GCN** | $77.3 \pm 0.9$ | $77.5 \pm 0.8$ | $61.1\pm0.9$ | $56.2\pm0.4$ | $77.6\pm0.9$ | $76.3\pm1.2$ |

levels, i.e., up to 99%. Instead of keeping the sparsity of $m$ fixed throughout the sparsification process, we start from an untrained dense GNNs and gradually increase the sparsity to the target sparsity during the whole sparsification process. We adjust the original gradual sparsification schedule [18, 19] to the linear decay schedule, since no big performance difference can be observed. The sparsity level $s_t$ of each adjusting step $t$ is calculated as follows:

$$s_t = s_f + (s_i - s_f)(1 - \frac{t - t_0}{n\Delta t}) \tag{5}$$
$$t \in \{t_0, t_0 + \Delta t, ..., t_0 + n\Delta t\}$$

where $s_f$ and $s_i$ refer to the final sparsity and initial sparsity, respectively; The initial sparsity is the sparsity at the start point of sparsification and it is set to 0 in this study. The final sparsity is the sparsity at the endpoint of sparsification. $t_0$ is the starting point of sparsification; $\Delta t$ is the time between two adjusting steps; $n$ is the total number of adjusting steps. We set $\Delta t$ as one epoch of mask optimization in this paper.

To obtain a good non-uniform layer-wise sparsity ratio, we remove the weights with the smallest score values ($S$) across layers at each adjusting step. We do this because [20] showed that the layer-wise sparsity obtained by this scheme outperforms the other well-studied sparsity ratios [19, 37, 39]. More importantly, removing weights across layers theoretically has a larger search space than solely considering one layer. The former can be more appealing as the GNN architecture goes deeper.

## 4 Experimental Results

In this section, we conduct extensive experiments among multiple GNN architectures and datasets to evaluate UGTs. We summarize the experimental setups here.

**Table 2:** Graph datasets statistics.

| DataSets | #Graphs | #Nodes | #Edges | #Classes | #Features | Metric |
|---|---|---|---|---|---|---|
| Cora | 1 | 2708 | 5429 | 7 | 1433 | Accuracy |
| Citeseer | 1 | 3327 | 4732 | 6 | 3703 | Accuracy |
| Pubmed | 1 | 19717 | 44338 | 3 | 3288 | Accuracy |
| OGBN-Arxiv | 1 | 169343 | 1166243 | 40 | 128 | Accuracy |
| Texas | 1 | 183 | 309 | 5 | 1703 | Accuracy |
| OGBN-Products | 1 | 24449029 | 61859140 | 47 | 100 | Accuracy |
| OGBG-molhiv | 41127 | 25.5(Average) | 27.5(Average) | 2 | - | ROC-AUC |
| OGBG-molbace | 1513 | 34.1(Average) | 36.9(Average) | 2 | - | ROC-AUC |

**GNN Architectures**. We use the three most widely used GNN architectures: GCN, GIN, and GAT [1] in our paper.

**Datasets**. We choose three popular small-scale graph datasets including Cora, Citeseer, PubMed [3] and one latest large-scale graph dataset OGBN-Arxiv [15] for our main experiments. To draw a

---

[1]All experiments based on GAT architecture are conducted with heads=1 in this study.

solid conclusion, we also evaluate our method on other datasets including OGBN-Products [15], TEXAS [43], OGBG-molhiv [15] and OGBG-molbace [15, 44]. More detailed information can be found in Table 2.

## 4.1 The Existence of Matching Subnetworks

Figure 1 shows the effectiveness of UGTs with different GNNs, including GCN, GIN and GAT, on the four datasets. We can observe that as the model size increases, UGTs can find untrained subnetworks that match the fully-trained dense GNNs. This observation is perfectly in line with the previous findings [12, 13], which reveal that model size plays a crucial role to the existence of matching untrained subnetworks. Besides, it can be observed that the proposed UGTs consistently outperforms Edge-Popup across different settings.

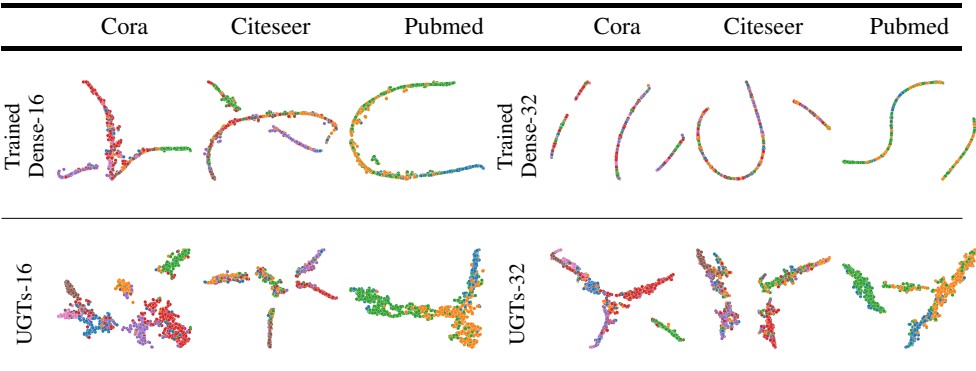

**Figure 3:** TSNE visualization of node representations learned by densely trained GCN and UGTs. Ten classes are randomly sampled from OGBN-Arxiv for visualization. Model depth is set as 16 and 32 respectively; width is set as 448. See Appendix B.1 for GAT architecture.

## 4.2 Over-smoothing Analysis

Deep architecture has been shown as a key factor that improves the model capability in computer vision [45]. However, it becomes less appealing in GNNs mainly because the node interaction through the message-passing mechanism (i.e., aggregation operator) would make node representations less distinguishable [26, 46], leading to a drastic drop of task performance. This phenomenon is well known as the over-smoothing problem [14, 42]. In this paper, we show a surprising result that UGTs can effectively mitigate over-smoothing in deep GNNs. We conduct extensive experiments to evaluate this claim in this section.

**UGTs preserves the high accuracy as GNNs go deeper.** In Figure 2, we vary the model depth of various architectures and report the test accuracy. All the experiments are conducted with architectures containing width 448 except for GAT on OGBN-Arxiv, in which we choose width 256 for GAT with $2 \sim 10$ layers and width 128 for GAT with $11 \sim 20$ layers, due to the memory limitation.

As we can see, the performance of trained dense GNNs suffers from a sharp performance drop when the model goes deeper, whereas UGTs impressively preserves the high accuracy across models. Especially at the mild sparsity, i.e., 0.1, UGTs almost has no deterioration with the increased number of layers.

**UGTs achieves competitive performance with the well-versed training techniques.** To further validate the effectiveness of UGTs in mitigating over-smoothing, we compare UGTs with six state-of-the-art techniques for the over-smoothing problem, including Residual connections, Jumping connections, NodeNorm, PairNorm, DropEdge, and DropNode. We follow the experimental setting in [42] and conduct experiments on Cora/Citeseer/Pubmed with GAT containing 16 and 32 layers. Model width is set to 448 for GAT on Cora/Citeseer/Pubmed. The results of the other methods are obtained from [42][2].

---

[2]https://github.com/VITA-Group/Deep_GCN_Benchmarking.git

Table 1 shows that UGTs consistently outperforms all these advanced techniques on Cora, Citeseer, and Pubmed. For instance, UGTs outperforms the best performing technique (+Jumping) by 2.0%, 1.2%, 1.0% on Cora, Citeseer and Pubmed respectively with 32 layers. These results again verify our hypothesis that training bottlenecks of deep GNNs (e.g., over-smoothing) can be avoided or mitigated by finding untrained subnetworks without training weights at all.

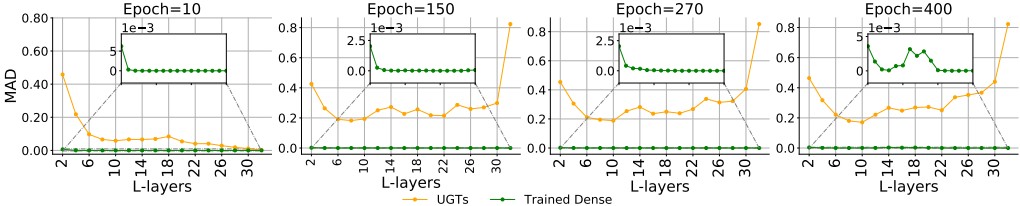

**Figure 4:** Mean Average Distance among node representations of each GNN layer. Experiments are conducted on Cora with GCN containing 32 layers and width 448.

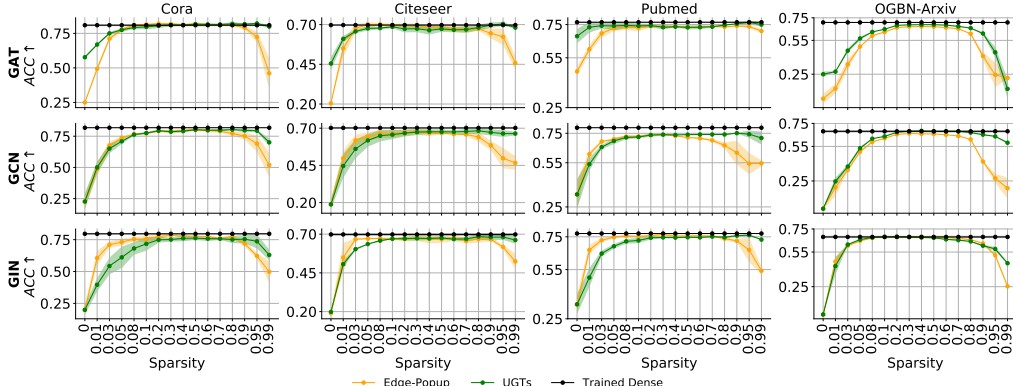

**Figure 5:** The accuracy of GNNs w.r.t varying sparsities. Experiments are conducted on various GNNs with 2 layers and width 256 for Cora, Citeseer and Pubmed, 4 layers and width 386 for OGBN-Arxiv.

**Mean Average Distance (MAD).** To further evaluate whether or not the good performance of UGTs can be contributed to the mitigation of over-smoothing, we visualize the smoothness of the node representations learned by UGTs and trained dense GNNs respectively. Following [46], we calculate the MAD distance among node representations for each layer during the process of sparsification. Concretely, MAD [46] is the quantitative metric for measuring the smoothness of the node representations. The smaller the MAD is, the smoother the node representations are. Results are reported in Figure 4. It can be observed that the node representations learned by UGTs keeps having a large distance throughout the optimization process, indicating a relieving of over-smoothing. On the contrary, the densely trained GCN suffers from severely indistinguishable representations of nodes.

**TSNE Visualizations.** Additionally, we visualize the node representations learned by UGTs and the trained dense GNNs with 16 and 32 layers, respectively, on both GCN and GAT architectures. Due to the limited space, we show the results of GCN in Figure 3 and put the visualization of GAT in the Appendix B.1. We can see that the node representations learned by the trained dense GCN are over-mixing in all scenarios and, in the deeper models (i.e., 32 layers), seem to be more indistinguishable. Meanwhile, the projection of node representations learned by UGTs maintains clearly distinguishable, again providing the empirical evidence of UGTs in mitigating over-smoothing problem.

## 4.3 The Effect of Sparsity on UGTs

To better understand the effect of sparsity on the performance of UGTs, we provide a comprehensive study in Figure 5 where the performance of UGTs with respect to different sparsity levels on different architectures. We summarize our observations below.

① **UGTs consistently finds matching untrained graph subnetworks at a large range of sparsities, including the extreme ones.** A matching untrained graph subnetwork can be identified with sparsities from 0.1 even up to 0.99 on small-scale datasets such as Cora, Citeseer and Pubmed. For large-scale OGBN-Arxiv, it is more difficult to find matching untrained subnetworks. Matching subnetworks are mainly located within sparsities of $0.3 \sim 0.6$.

② **What's more, UGTs consistently outperforms Edge-Popup.** UGTs shows better performance than Edge-Popup at high sparsities across different architectures on Cora, Citeseer, Pubmed and OGBN-Arxiv. Surprisingly, increasing sparsity from 0.7 to 0.99, UGTs maintains very a high accuracy, whereas the accuracy of Edge-Popup shows a notable degradation. It is in accord with our expectation since UGTs finds important weights globally by searching for the well-performing sparse topology across layers.

## 4.4 Broader Evaluation of UGTs

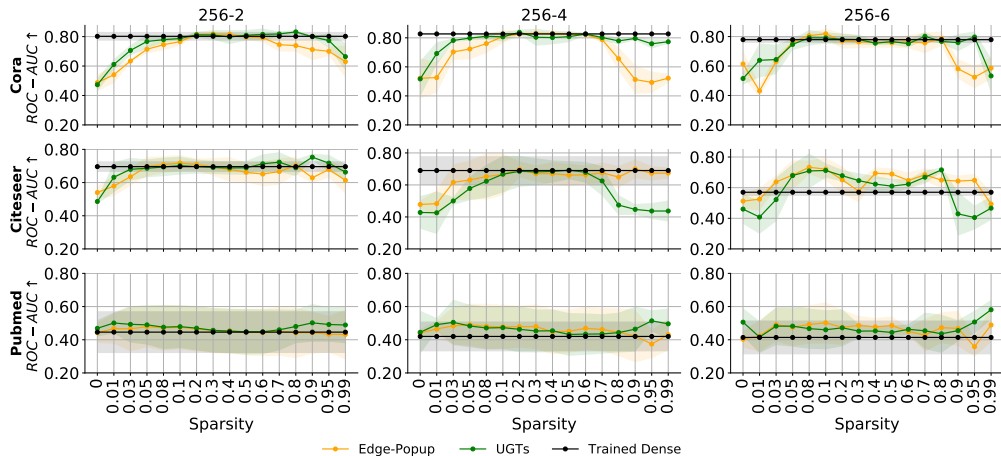

**Figure 6:** Out-of-distribution performance (ROC-AUC). Experiments are conducted with GCN (Width: 256, Depth: 2).

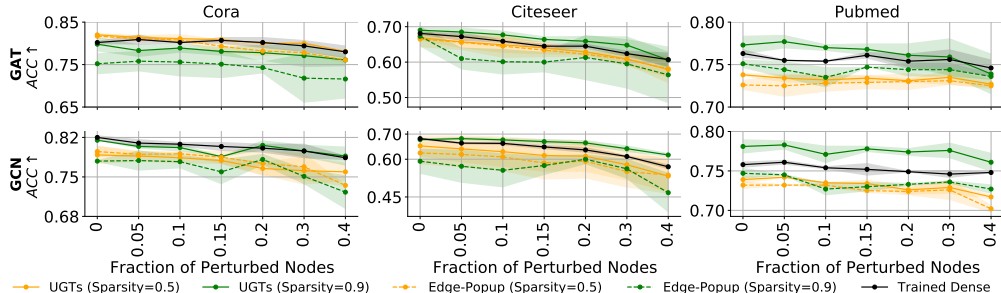

**Figure 7:** The robust performance on feature perturbations with the fraction of perturbed nodes varying from 0% to 40%. Experiments are conducted with GCN and GAT (Width: 256, Depth: 2).

In this section, we systematically study the performance of UGTs on out of distribution (OOD) detection, robustness against the input perturbations including feature and edge perturbations. Following [47], we create OOD samples by specifying all samples from 40% of classes and removing them from the training set. We create feature perturbations by replacing them with the noise sampled from Bernoulli distribution with p=0.5 and edge perturbations by moving edge's end point at random. The results of OOD experiments are reported in Figure 6 and Figure 10 (shown in Appendix B.2). The results of robustness experiments are reported in Figure 8 and Figure 7. We summarize our observations as follows:

① **UGTs enjoys matching performance on OOD detection.** Figure 6 and Figure 10 show that untrained graph subnetworks discovered by UGTs achieve matching performance on OOD detection

compared with the trained dense GNNs in most cases. Besides, UGTs consistently outperforms Edge-Popup method at a large range of sparsities on OOD detection.

② **UGTs produces highly sparse yet robust subnetworks on input perturbations.** Figure 7 and Figure 8 demonstrate that UGTs with high sparsity level (Sparsity=0.9) achieves more robust results than the trained dense GNNs on both feature and edge perturbations with perturbation percentage ranging from 0 to 40%. Again, UGTs consistently outperforms Edge-Popup with both perturbation types.

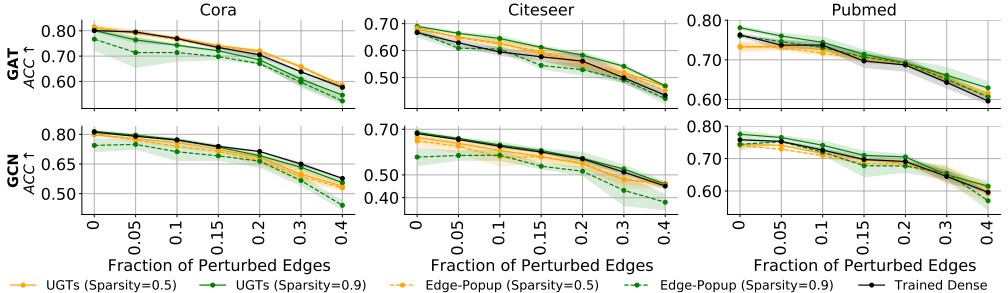

**Figure 8:** The robust performance on edge perturbations with the fraction of perturbed edges varying from 0% to 40%. Experiments are conducted with GCN and GAT (Width: 256, Depth: 2).

**Table 3:** Experments on graph-level tasks and other datasets. GCN Model with width:448, depth:3 are adopted for this experiments.

| Node-level: OGBN-Products (Accuracy:%) | | | | | | | | | | | |
|---|---|---|---|---|---|---|---|---|---|---|---|
| Sparsity= | 0.1 | 0.2 | 0.3 | 0.4 | 0.5 | 0.6 | 0.7 | 0.8 | 0.9 | 0.95 | 0.99 |
| Dense | 79.5 | 79.5 | 79.5 | 79.5 | 79.5 | 79.5 | 79.5 | 79.5 | 79.5 | 79.5 | 79.5 |
| **UGTs** | 75.6 | 77.7 | 78.7 | 77.8 | 79.3 | 79.5 | 79.9 | 78.6 | 74.5 | 64.1 | 35.3 |
| Node-level: TEXAS (Accuracy:%) | | | | | | | | | | | |
| Dense | 62.2 | 62.2 | 62.2 | 62.2 | 62.2 | 62.2 | 62.2 | 62.2 | 62.2 | 62.2 | 62.2 |
| **UGTs** | 62.1 | 62.2 | 62.2 | 62.2 | 62.2 | 61.3 | 62.2 | 63.1 | 64.8 | 64.8 | 55.8 |
| Graph-level: OGBG-molhiv (ROCAUC:%) | | | | | | | | | | | |
| Dense | 77.4 | 77.4 | 77.4 | 77.4 | 77.4 | 77.4 | 77.4 | 77.4 | 77.4 | 77.4 | 77.4 |
| **UGTs** | 76.4 | 76.9 | 76.5 | 76.1 | 76.3 | 77.3 | 75.8 | 77.1 | 73.1 | 75.3 | 75.1 |
| Graph-level: OGBG-molbace (ROCAUC:%) | | | | | | | | | | | |
| Dense | 78.3 | 78.3 | 78.3 | 78.3 | 78.3 | 78.3 | 78.3 | 78.3 | 78.3 | 78.3 | 78.3 |
| **UGTs** | 76.0 | 73.7 | 77.0 | 77.1 | 77.0 | 77.6 | 78.4 | 77.3 | 76.2 | 75.6 | 74.9 |

## 4.5 Experiments on Graph-level Task and Other Datasets

To draw a solid conclusion, we further conduct extensive experiments of graph-level task on OGBG-molhiv and OGBG-molbace; node-level task on TEXAS and OGBN-Products. The experiments are based on GCN model with width=448 and depth=3. Table 3 consistently verifies that a matching untrained subnetwork can be identified in GNNs across multiple tasks and datasets.

## 5 Conclusion

In this work, we for the first time confirm the existence of matching untrained subnetworks at a large range of sparsity. UGTs consistently outperforms the previous untrained technique – Edge-Popup on multiple graph datasets across various GNN architectures. What's more, we show a surprising result that searching for an untrained subnetwork within a randomly weighted dense GNN instead of directly training the latter can significantly mitigate the over-smoothing problem of deep GNNs. Across popular datasets, e.g., Cora, Citeseer, Pubmed, and OGBN-Arxiv, our method UGTs can achieve comparable or better performance with the various well-studied techniques that are specifically designed for over-smoothing. Moreover, we empirically find that UGTs also achieves appealing performance on other desirable aspects, such as out-of-distribution detection and robustness. The

strong results of our paper point out a surprising but perhaps worth-a-try direction to obtain high-performing GNNs, i.e., finding the Untrained Tickets located within a randomly weighted dense GNN instead of training it.

## Acknowledgements

This work used the Dutch national e-infrastructure with the support of the SURF Cooperative using grant no. NWO-2021.060 and EINF-3214/L1.

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

# A    Implementation Details

In this paper, all experiments on Cora/Citeseer/Pubmed datasets are conducted on 1 GeForce RTX 2080TI (11GB) and all experiments on OGBN-Arxiv are conducted on 1 DGX-A100 (40GB). All the results reported in this paper are conducted by 5 independent repeated runs.

**Train-Val-Test splitting Datasets** We use 140 (Cora), 120 (Citeseer) and 60 (PubMed) labeled data for training, 500 nodes for validation and 1000 nodes for testing. We follow the strategy in [15] for splitting OGBN-Arxiv dataset.

**Hyper-parameter Configuration** We follow [3, 42, 48] to configure the hyper-parameters for training dense GNN models. All hyper-parameters configurations for UGTs are summarized in Table 4.

**Table 4:** Implementation details for UGTs.

| DataSets | Cora | Citeseer | Pubmed | OGBN-Arxiv |
|---|---|---|---|---|
| Total Epoches | 400 | 400 | 400 | 400 |
| Learning Rate | 0.01 | 0.01 | 0.01 | 0.01 (GNNs with Layers<10) 0.001(GNNs with Layers>10) |
| Optimizer | Adam | Adam | Adam | Adam |
| Weight Decay | 0.0 | 0.0 | 0.0 | 0.0 |
| n(total adjustion epoches) | 200 | 200 | 200 | 200 |
| $t_0$ | 0 | 0 | 0 | 0 |
| $\Delta t$ | 1 epoch | 1 epoch | 1 epoch | 1 epoch |

# B    More Experimental Results

## B.1    TSNE visualization.

Figure 9 provides the TSNE visualization for node representations learned by UGTs and dense GAT. It can be observed that the node representations learned by the trained dense GAT are mixed while the node representations learned by UGTs are disentangled.

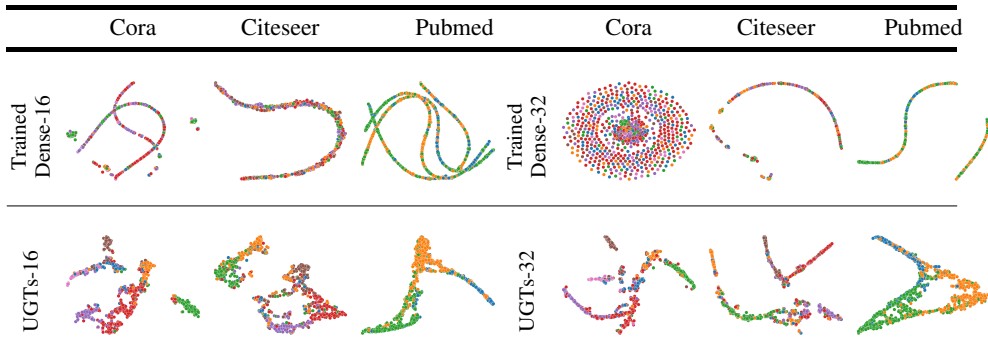

**Figure 9:** TSNE visualization for node representations. Experiments are based on GAT with fixed width 448.

## B.2    Out of distribution detection

Figure 10 shows the OOD performance for UGTs and the trained dense GNNs based on GAT architecture. As we can observe, UGTs achieves very appealing results on OOD performance than the corresponding trained dense GAT.

## B.3    Robustness against input perturbations

In this section, we explore the robustness against input perturbations with varying the sparsity of untrained GNNs. Experiments are conducted on GAT and GCN architectures with width=256 and depth=2. Results are reported in Figure 12 and Figure 11.

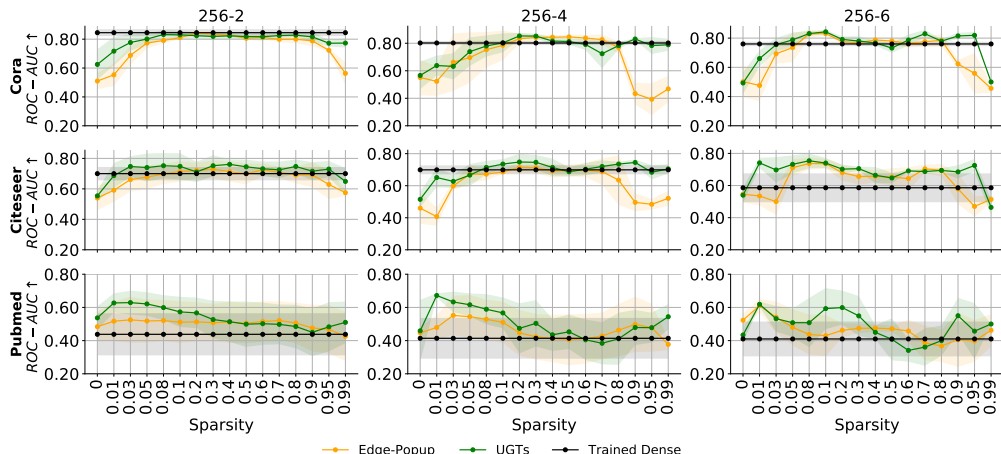

**Figure 10: Out-of-distribution performance (ROC-AUC).** Experiments are based on GAT architecture (Width:256, Depth:2)

It can be observed that the robustness achieved by UGTs is increasing with the increase of sparsity for both edge and feature perturbation types. Besides, the robustness achieved by UGTs at large sparsity, e.g., sparsity =0.9, can outperform the counterpart trained dense GNNs.

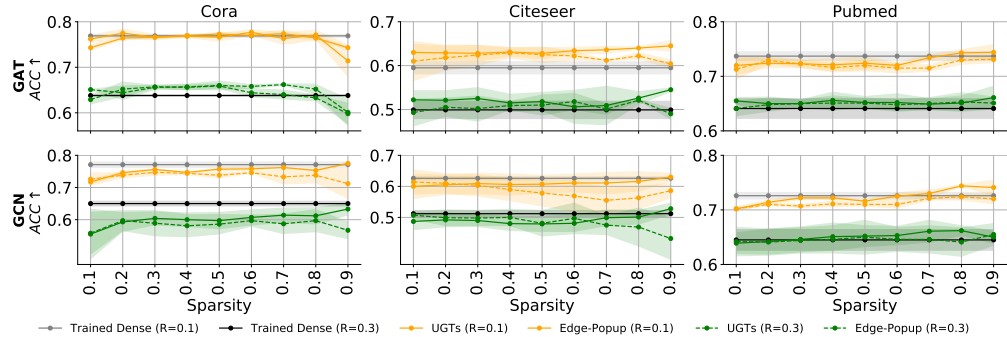

**Figure 11: The robust performance on edge perturbations.** $R$ denotes the fraction of perturbed edges.(width:256, Depth:2)

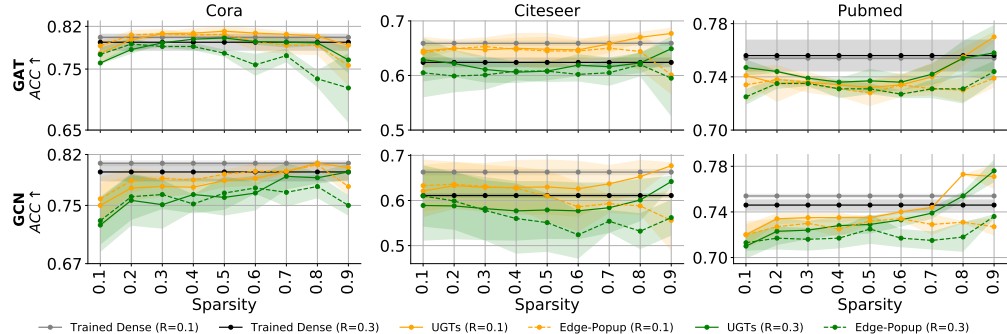

**Figure 12: The robust performance on feature perturbations.** $R$ denotes the fraction of perturbed nodes. (width:256, Depth:2)

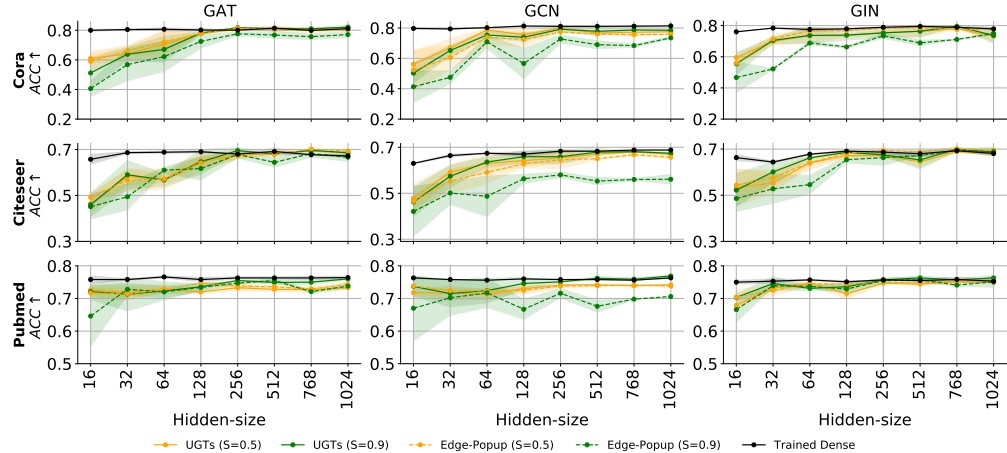

**Figure 13: Model Width:** The accuracy performance of subnetworks from untrained GNNs w.r.t varying Hidden-Size. The "S01,S05,S09" represents the sparsity of the untrained GNNs. The dashed line represents the results of the trained dense GNNs. Experiments are based on GNNs with 2 layers.

## B.4   The accuracy performance w.r.t model width

Figure 13 shows the performance of UGTs on different architectures with varying model width from 16 to 1024 and fix depth=2. We summarize observations as follows:

① **Performance of UGTs improves with the width of the GNN models.** With width increasing from 16 to 256, the performance of UGTs improves apparently and after width=256, the benefits from model width are saturated.

## B.5   Ablation studies

We conduct the ablation studies to show the effectiveness of UGTs. The results showed on Table 5. Compared with Edge-Popup, UGTs mainly has two novelties: global sparsification (VS. uniform sparsification) and gradual sparsification (VS. one-shot sparsification). Here we compare UGTs with 3 baselines: (1) UGTs - global sparsification; (2) UGTs - gradual sparsification; (3) Edge-Popup. The results are reported in the following table and it shows that global sparsification plays an important role for finding important weights and gradual sparsification is crucial for further boosting performance at high sparsity level.

**Table 5:** Ablation studies based on GAT (Depth:2, Width:256) and Cora.

|  | 0.1 | 0.3 | 0.5 | 0.7 | 0.9 | 0.95 |
|---|---|---|---|---|---|---|
| Edge-Popup | **0.814** | 0.81 | 0.809 | 0.81 | 0.791 | 0.461 |
| UGTs - global sparsification | 0.807 | 0.816 | 0.817 | 0.804 | 0.799 | 0.731 |
| UGTs - gradual sparsification | 0.806 | **0.818** | **0.821** | **0.821** | 0.804 | 0.795 |
| UGTs | 0.797 | 0.811 | 0.81 | 0.815 | **0.817** | **0.822** |

## B.6   Observations via gradient norm

To preliminary understand why UGTs can mitigate over-smoothing while the trained dense GNNs can not, we calculate the gradient norm of each layer for UGTs and dense GCN during training. In order to have a fair comparison, we calculate the gradient norm of $\nabla_{(\boldsymbol{m}^l \odot \boldsymbol{\theta}^l)} \mathcal{L}(g(\boldsymbol{A}, \boldsymbol{X}; \boldsymbol{\theta} \odot \boldsymbol{m}), y)$ for UGTs and the gradient norm of $\nabla_{\boldsymbol{\theta}^l} \mathcal{L}(g(\boldsymbol{A}, \boldsymbol{X}; \boldsymbol{\theta}), y)$ for dense GCN where $l$ denotes the layer. Results are reported in Figure 14.

As we can observe, the gradient vanishing problem may exist for training deep dense GCN since the gradient norm for dense GCN is extremely small while UGTs does not have this problem. This problem might also be indicated by the training loss where the training loss for dense GCN does not

decrease while the training loss for UGTs decreases a lot. This might explain why UGTs performs well for deep GNNs.

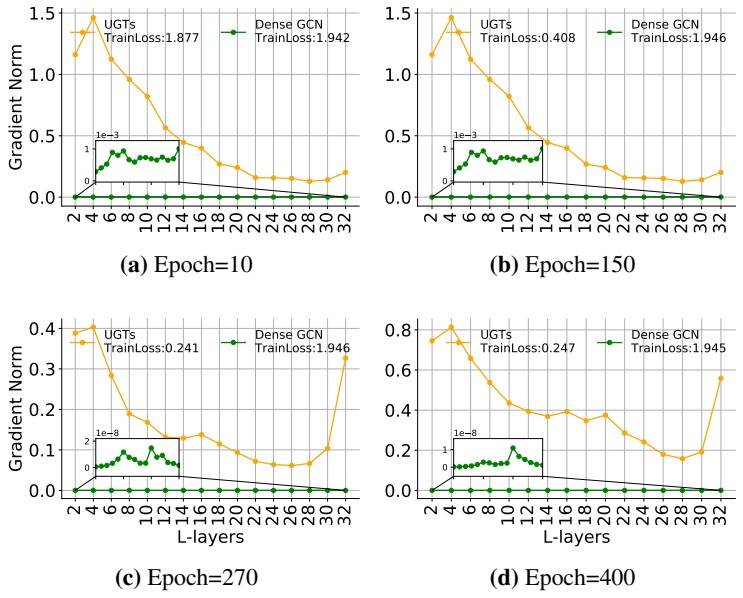

**Figure 14: Gradient norm w.r.t each layer during training.** Experiments are conducted on Cora with GCN architecture containing 32 layers and width 448.

## B.7 More experiments for mitigating over-smoothing problem

We conduct experiments on Cora, Citeseer and Pubmed for GAT with deeper layers. The width is fixed to 448. The results of the other methods are obtained by running the code[3].

The results are reported in Table 6. It can be observed again that UGTs consistently outperforms all the baselines.

**Table 6:** Test accuracy (%) of different training techniques. The experiments are based on GAT models with 16, 32 layers, respectively. Width is set to 448.

|  | Cora | | Citeseer | | Pubmed | |
|---|---|---|---|---|---|---|
| N-Layers | 16 | 32 | 16 | 32 | 16 | 32 |
| **Trained Dense GAT** | 20.6 | 13.0 | 20.0 | 16.9 | 17.9 | 18.0 |
| +Residual | 19.9 | 20.7 | 17.7 | 19.2 | 41.6 | 40.8 |
| +Jumping | 39.7 | 27.8 | 29.1 | 25.5 | 57.3 | 57.1 |
| +NodeNorm | 70.9 | 11.0 | 17.1 | 18.4 | 72.2 | 59.7 |
| +PairNorm | 27.9 | 12.1 | 22.8 | 17.7 | 73.0 | 44.0 |
| +DropNode | 23.6 | 13.0 | 18.8 | 7.0 | 26.7 | 18.0 |
| +DropEdge | 24.8 | 13.0 | 19.4 | 7.0 | 19.3 | 18.0 |
| **UGTs-GAT** | **76.7 ± 1.1** | **74.9±0.2** | **62.7±0.7** | **56.5±1.1** | **77.9±0.5** | **75.5±1.5** |

## C Pseudocode

Pseudocode is showed in Algothrim 1.

---

[3] https://github.com/VITA-Group/Deep_GCN_Benchmarking.git

---

**Algorithm 1** Untrained GNNs Tickets (UGTs)

---

**Input:** a GNN $g(\boldsymbol{A}, \boldsymbol{X}; \boldsymbol{\theta})$, initial mask $\boldsymbol{m} = 1 \in \mathcal{R}^{|\boldsymbol{\theta}|}$ with latent scores $\boldsymbol{S}$, learning rate $\lambda$, hyperparameters for the gradual sparsification schedule $s_i$, $s_f$, $t_0$, and $\Delta t$.
**Output:** $g(\boldsymbol{A}, \boldsymbol{X}; \boldsymbol{\theta} \odot \boldsymbol{m}), \boldsymbol{y}$
Randomly initialize model weights $\boldsymbol{\theta}$ and $\boldsymbol{S}$.
**for** $t = 1$ **to** $T$ **do**
    #Calculate the current sparsity level $s_t$ by Eq. 5.
    $s_t \longleftarrow s_f + (s_i - s_f)(1 - \frac{t - t_0}{n \Delta t})$
    #Get the global threshold value at top $s_t$ by sorting $\boldsymbol{S}$ in ascending order.
    $\boldsymbol{S}_{thres} \longleftarrow Thresholding(\boldsymbol{S}, s_t)$
    #Generate the binary mask.
    $\boldsymbol{m} \longleftarrow 0$ if $\boldsymbol{S} < \boldsymbol{S}_{thres}$ else $1$
    #Update $\boldsymbol{S}$.
    $\boldsymbol{S} \longleftarrow \boldsymbol{S} - \lambda \nabla_{\boldsymbol{S}} \mathcal{L}(g(A, \boldsymbol{X}; \boldsymbol{\theta} \odot \boldsymbol{m}), \boldsymbol{y})$
**end for**
Return $g(\boldsymbol{A}, \boldsymbol{X}; \boldsymbol{\theta} \odot \boldsymbol{m}), \boldsymbol{y}$

---

