# OpenReview forum: "You Can Have Better Graph Neural Networks by Not Training Weights at All: Finding Untrained GNNs Tickets"
_logconference.io/LOG/2022/Conference — LoG 2022 Oral_

### Official Review · Reviewer_HVMt · 2022-09-22

**Overall Score:** 6
**Confidence:** 4

**Review:**

This paper explores the concept of pruning a GNN with masks and evaluation of the selected sub-graph without training the weights. It is an interesting subject, which is the intersection of pruning and learning with random weights. I have the following comments:
- How the latent score is calculated after (3)?
- How the top-s largest elements are selected (i.e. the threshold)?
- \theta is not defined (I assume it refers to the weights).
- Is m a vector in (3)?
- In (5), how s_f and s_i are defined?
- Why the sparse untrained graph is equivalent to a dense network? How the capacity of dense network is calculated? What kind of dense network? Should there be a reference dense network?
In overall, the paper has provided experiments to explore the concept, but the concept needs more development and results should be discussed in more details. The flow of equations and method (which is basically masking weights) needs to be improved for better readability of the paper.

The paper has 9 pages + appendix.

---

### Official Review · Reviewer_5K6D · 2022-10-18

**Overall Score:** 8
**Confidence:** 4

**Review:**

**Contributions**

Recent research has shown that it is possible to find a sparse subnetwork with random weights in a convolutional neural network, such that the performance of the subnetwork matches that of the dense, trained one.
This work extends this idea to graph neural networks.

The core idea presented in the paper is straightforward and the contributions of the work are primarily experimental. The main contributions of the paper are:

1. Adapting existing algorithms for finding sparse subnetworks to the GNN case. The authors propose minor changes to previous techniques that allow them to push the sparsity of the subnetworks to up to 99%.
2. Showing that the subnetworks perform on par with the dense trained networks on typical node and graph classification benchmarks.
3. Showing that the sparse subnetworks solve the over-smoothing problem of naive GNNs, allowing to design architectures with up to 32 layers. The authors also show that other approaches to solve oversmoothing are less performant than the sparse subnetworks found by the proposed method.
4. Showing that the sparse subnetworks exhibit good generalization to out-of-distribution data and are also robust to graph perturbations.

**Strengths**

1. The paper shows interesting results about (not) training graph neural networks that, I believe, could be of significant interest to the larger GNN community.
2. The experimental analysis is thorough and does not leave many questions unanswered (though please see below).
3. The algorithm is easy to understand, and it should be possible to implement/reproduce results easily.
4. The paper is easy to read, and is well-contextualized within previous literature.

**Weaknesses**

1. It is not clear if the proposed method is cheaper than training a dense network as usual. I suggest the authors report the training times of the base GNN compared to the time required to find the mask with their method.
2. The explanation of why EdgePopup performs comparably to the proposed method when using GIN is not really plausible. Why would a larger search space affect the model differently when using GIN rather than GCN/GAT? I suggest to remove that observation altogether, since it would not be easy to support. It would be a valuable addition to the paper if the authors found theoretical or empirical evidence for why this happens.
3. Depending on the conference's standard for novelty, the core contributions of the paper could be perceived as minor. For example, nothing in the proposed idea is specific to GNNs (in fact, an interesting question would be if the proposed linear schedule for the sparsity can improve performance also in CNNs).

**Recommendation**

The paper is interesting and the weaknesses are minor. I think this paper could be a valuable contribution to LoG and the GNN community.

Therefore, I can safely recommend acceptance, although the authors should consider the comments above (and the additional feedback) to improve the paper.

**Questions**

1. Figure 4 shows something is clearly going on when the depth exceeds 30 layers. However, the accuracy reported in Figure 2 goes up to 20 layers. What is the relation between the spike in MAD and the task accuracy?

**Additional feedback**

- The acronyms for LTH and OOD are never introduced.
- Figure 1, caption: "64 widths" -> "64 units" (or: "width 64").
- Line 97: "it is consisted" -> "it consists"
- Line 134: "layer-wisely" -> "layer-wise"
- The expression "untrained graph tickets" does not really express the idea of the paper; the ticket is a subset of a dense matrix of weights, which has little to do with the input graph. A more precise expression would be "untrained tickets in GNNs."
- The experimental results on depth would be more interesting if the authors considered a dataset(s) where deeper networks exhibit better performance. There, the advantages of UGTs would probably be more evident.

---

### Official Review · Reviewer_vyWy · 2022-10-19

**Overall Score:** 8
**Confidence:** 3

**Review:**

This paper focuses on an interesting problem, i.e., finding an untrained graph subnetwork for DNN which has a comparable performance with trained GNN. The paper proposes a simple yet effective method UGT, which considers different sparsity levels for each step. The paper also provides an interesting observation that the proposed method can mitigate the over-smoothing problem in conventional deep GNNs. The extensive experiments demonstrate its observation. The experimental results also show that the proposed UGT outperforms the conventional method Edge-Popup. The topic is interesting, and the experiments are convincing and show the effectiveness of the proposed method.

I have the following questions and comments:
1. Although the authors claim that the goal is not for efficiency, I'm still wondering about its efficiency. Could the authors provide the running time results in the experiments? If it takes more time for UGT to optimize Eq,(4) than fully training a GNN, is this task still meaningful or is this method still useful?
2. The paper designs a new strategy to set different sparsity. It would be better to conduct some ablation study in experiments to show the effectiveness of the strategy.

---

### Official Review · Reviewer_MZfR · 2022-10-22

**Overall Score:** 8
**Confidence:** 5

**Review:**

### Summary
The authors explore untrained GNNs. The idea is inspired by similar results on CNNs. They show that one can find untrained sparse subnetworks at the initialization, matching the performance of fully trained dense GNNs. A byproduct of this exploration is that these untrained networks can deal with the oversmoothing problem while having stability against input perturbations and generalizing capability to out-of-distribution detection. Their proposed approach -- untrained graph tickets, discovers untrained matching subnetworks at very high sparsities. The results are empirically demonstrated on well-known benchmark datasets.

### Strengths
1. This work tackles a well-motivated problem. It is interesting to see that discovering untrained subnetworks is also possible for GNNs. Although this is not surprising to me because it's done for CNNs already, it's still very interesting.
2. The paper is extremely well-written and easy to follow. Intuitive and clear explanations are provided throughout the paper for every decision made.
3. A very thorough set of experiments have been conducted to demonstrate the effectiveness of the proposed approach. The findings are compelling.
4. I enjoyed reading the paper. The previous literature is also well summarized.

### Weaknesses
1. I found the description of "sparsity" a bit confusing at the beginning (perhaps because I'm not too familiar with this literature). Especially in line 56, I believe the authors could say something more about what it means to say "at extremely high sparsities". This becomes clear to me later when I look at the equations and the pseudocode, but some intuition early on would be helpful.
2. The oversmoothing analysis is very interesting, however, it is purely empirical. It'd be nice to see some kind of theoretical result that shows how UGTs prevent the node features from becoming indistinguishable as more layers of a GNN are piled up.

### Questions
1. Do the authors have a direction for some theoretical analysis/result that shows how UGTs prevent oversmoothing (node features becoming indistinguishable)?
2. Could you please explain lines 147 -- 151 a bit more? Perhaps an intuition for why removing weights across layers has a larger search space would be good (maybe this is obvious, but not to me).
3. In line 236, does "specifying all samples from 40% of classes and removing them from the training set" mean that the training sample consists of only data from 60% of classes? If so, then for the test set, what is the ratio of data points in the classes that were present in the training sample to classes that were absent? Is this ratio also 2:3? I ask this to understand what the accuracy metric truly represents for the out-of-distribution detection plots.
4. In line 237, you say that "feature perturbations by adding noise from Bernoulli distribution". What does this exactly mean? Do we change the feature to $0/1$ with some probability? Or do we add $0/1$ to each coordinate in the feature vector with some probability? Please clarify in the paper if possible.
5. What kind of mathematical tools/frameworks are required to provide theoretical guarantees about the results that are observed empirically? Especially, the mitigation of the oversmoothing problem and robustness to perturbations seem to be theoretically doable. Is there any existing literature for CNNs that provides such guarantees? Could that be adopted for GNNs as well?
6. **[Important]** In the pseudocode for UGT in appendix C, what is the function Thresholding$({\bf S}, s_t)$ doing exactly?

---

### Meta-Review · Area_Chair_vpKN · 2022-11-12

**Confidence:** 5
**Recommendation:** Accept for spotlight

**Meta Review:**

The authors tackle the quest for lottery tickets for GNNs. The findings are insightful, and likely to be extremely useful to the community. After an extensive discussion between the authors and the reviewers, a consensus on this has been reached by all reviewers.

Personally I wholeheartedly agree that the findings of this paper are highly important, and given that it is among the top-rated papers of this year's LoG, I would recommend it for a contributed talk.

---

### Decision · Program_Chairs · 2022-11-23

Accept (Oral)